# Myelopoiesis during Solid Cancers and Strategies for Immunotherapy

**DOI:** 10.3390/cells10050968

**Published:** 2021-04-21

**Authors:** Tyler J. Wildes, Bayli DiVita Dean, Catherine T. Flores

**Affiliations:** University of Florida Brain Tumor Immunotherapy Program, Preston A. Wells, Jr. Center for Brain Tumor Therapy, Lillian S. Wells Department of Neurosurgery, McKnight Brain Institute, University of Florida, Gainesville, FL 32607, USA; twildes@ufl.edu (T.J.W.); bayli.divitadean@neurosurgery.ufl.edu (B.D.D.)

**Keywords:** myelopoiesis, hematopoiesis, hematopoietic stem and progenitor cell, myeloid progenitor, solid cancer, malignancy, immunotherapy, immuno-oncology

## Abstract

Our understanding of the relationship between the immune system and cancers has undergone significant discovery recently. Immunotherapy with T cell therapies and checkpoint blockade has meaningfully changed the oncology landscape. While remarkable clinical advances in adaptive immunity are occurring, modulation of innate immunity has proven more difficult. The myeloid compartment, including macrophages, neutrophils, and dendritic cells, has a significant impact on the persistence or elimination of tumors. Myeloid cells, specifically in the tumor microenvironment, have direct contact with tumor tissue and coordinate with tumor-reactive T cells to either stimulate or antagonize cancer immunity. However, the myeloid compartment comprises a broad array of cells in various stages of development. In addition, hematopoietic stem and progenitor cells at various stages of myelopoiesis in distant sites undergo significant modulation by tumors. Understanding how tumors exert their influence on myeloid progenitors is critical to making clinically meaningful improvements in these pathways. Therefore, this review will cover recent developments in our understanding of how solid tumors modulate myelopoiesis to promote the formation of pro-tumor immature myeloid cells. Then, it will cover some of the potential avenues for capitalizing on these mechanisms to generate antitumor immunity.

## 1. Introduction

From an immunological perspective, solid cancers are thought to be the result of unbridled cell growth in an immunologically ignorant setting. Cancers are often able to evade the normal immune defenses because they are derived from self and therefore have very few differences in comparison to normal homeostatic tissue. Cancers can additionally turn on or off specific molecules to trick the immune system [1]. Some of the initial investigations on this interplay led to the development of early immunotherapy treatments. With that investigation, immunotherapy has dramatically enhanced our ability to understand the interplay between immunity and cancer and has provided a novel angle from which we can attack cancer [1,2,3,4]. While investigation in cancer immunotherapy has recently exploded, much of the native interaction between immunity and cancers is poorly understood.

Some of the known changes in native immunity in solid cancer patients include the proliferation of regulatory T cells [5], the generation of myeloid-derived suppressor cells [6,7,8], and the exhaustion of cytotoxic T cells [9]. These changes, at least in part, contribute to the state of immunosuppression often recognized in cancer patients. Most of the attention in the immunotherapy space has been given to novel checkpoint blockade monoclonal antibodies or T cell therapies, given their dramatic success in treating malignancies [10]. While coopting adaptive immunity is deserving of attention, there remains a large amount of untapped potential in the myeloid compartment, the arm of immunity that often decides and drives the initial response to immune targets.

There have been many recent advancements in our understanding of myeloid compartment interactions, specifically in the tumor microenvironment (TME). The work of Gabrilovich and colleagues led to the description of a whole new class of myeloid cells—the myeloid-derived suppressor cells (MDSCs) that are principally responsible for turning off immunity in the TME [6,7]. In addition, recent studies on the role of intratumoral CD103^+^ DCs and their requirement for T cell therapy efficacy have been described by multiple groups [11,12,13,14,15,16,17,18]. The initial intratumoral myeloid cell descriptions and subsequent studies by many other investigators have undoubtedly led to the recognition of dramatic consequences of the myeloid component of the TME in cancers ranging from melanoma to brain tumors. While the TME is fascinating and an area of intense investigation, we are also interested in the impact of solid cancers on peripheral myeloid cells, or myeloid cells not immediately involved or located in the TME. Taking that notion one step further, we were curious about how solid cancers might impact the generation of myeloid cells in the periphery, or myelopoiesis. In this area, there is considerably less understood.

Therefore, the purpose of this review is to delineate exactly what is known about how solid cancers impact native myelopoiesis. To achieve this, we will sometimes describe findings belonging to the broad category of hematopoietic stem and progenitor cells (HSPCs), a heterogeneous population of early myeloid progenitors and stem cells. Through understanding these details, perhaps we will have actionable components of native myelopoiesis that are targetable in our development of novel cancer immunotherapy agents.

## 2. Role of Effector Myeloid Cells in the Tumor Microenvironment

Before focusing on early myeloid cell development and function in the periphery, a brief discussion of the importance of effector myeloid cells in the TME is required. Ultimately, much of what occurs in the peripheral myelopoiesis will enact its end function in the TME and drive the clearance or persistence of tumors. Therefore, we will cover DCs and MDSCs briefly.

Dendritic cells in the context of modern cancer immunology have become an area of intense interest. In other reviews, DCs and their activities in the TME are covered in great detail [17,18]. In the past 12 years, investigation into intratumoral DCs has led to the realization that, for successful immune clearance of tumors, intratumoral DCs are required. Early studies by Merad and colleagues recognized the similarity between lymphoid-resident CD8^+^ DCs and TME-infiltrating CD103^+^ DCs and their common reliance on Batf3 [16]. Further studies described the reliance of successful checkpoint inhibitor therapies and BRAF inhibitors on CD103^+^ tumor-infiltrating DCs [11]. Significant following studies by Gajewski and colleagues led to the recognition that Batf3-driven CD103^+^ DCs are required for adoptive T cell therapy [12]. Those specific studies demonstrated that CD103^+^ DCs recruited T cells to the TME and were drivers of the inflamed TME that promotes tumor clearance.

MDSCs, a heterogenous population of immature myeloid cells with distinct immunosuppressive functions, comprise the majority of the immunosuppressive myeloid action in the tumor microenvironment. While MDSCs are derived from myeloid progenitors from the periphery, the majority of their action occurs in the TME. The definition of MDSCs has undergone significant clarification over time. Currently, phenotypic definitions are that mouse polymorphonuclear, or granulocytic MDSCs (gMDSCs), are CD11b^+^Ly6C^−^Ly6G^+^, and monocytic MDSCs (mMDSCs) are CD11b^+^Ly6C^+^Ly6G^−^ [19]. In humans, gMDSCs are CD11b^+^CD14^−^CD15^+^ and mMDSCs are CD11b^+^CD14^+^CD15^−^ [19]. Importantly, gMDSCs predominate in the periphery of solid tumor-bearing patients in sites such as the spleen, while mMDSCs predominate in the TME of solid tumors. In these semi-distinct settings, mMDSCs tend to have more functionally suppressive action in the TME as compared to gMDSCs [20,21]. While MDSCs and their actions are one of the primary results of dysregulated myelopoiesis, they have already been reviewed extensively [6,7].

## 3. Early Myeloid Cell Development

HSPC differentiation begins with the truly multipotent, self-renewing hematopoietic stem cell (HSCs) (Figure 1). While HSCs have been thought of as a simple stem cell, they possess considerable immune-modulating potential [22,23]. HSCs were originally thought to follow a linear path to differentiation; however, more recent studies have found mature effector cells are derived from a heterogenous pool of HSPCs [24]. In both mice and humans, these cells are generally characterized by their expression of CD34 and lack of expression of markers of mature immune cells, referred to as lineage.

Although some controversy remains on the exact mechanism by which HSCs undergo differentiation, most agree they next mature into multipotent progenitors, such as the short-term HSC (stHSC) and multipotent progenitor (MPP). MPPs play a crucial role in rescuing and reconstituting myeloablated hosts, and studies have shown lineage-restriction may occur in MPP subsets, with MPP2 and MPP3 possessing a preferential differentiation into myeloid cells [24,25,26]. In addition, using murine models, authors have shown MPPs are capable of differentiating into progenitors before undergoing cell division [27].

Downstream of MPPs are lineage-committed progenitors that split the lymphoid and myeloid compartments [25,28]. Common myeloid precursors (CMPs) give rise to mature myeloid cells, such as dendritic cells and macrophages, as well as red blood cells, whereas common lymphoid precursors (CLPs) will progress to become mature lymphoid cells, such as T cells, B cells, and NK cells. Multipotent progenitors continue to mature by differentiating into oligopotent, lineage-restricted progenitors, such as the megaryocytoke erythrocyte progenitor (MEP). MEPs are precursors to red blood cells and platelets; however, the role of these cells in cancer immunity is not well known.

Another example of an oligopotent progenitor is the monocyte dendritic cell precursor (MDP). As its name suggests, the cells can go on to become common DC precursors (CDPs) and common monocyte precursors (cMoP). CDPs differentiate into mature conventional and plasmacytoid DCs, where cMoPs become monocytes, monocyte-derived dendritic cells, monocyte-derived macrophages, or monocytic myeloid-derived suppressor cells (mMDSCs) [29,30].

A final example of an oligopotent progenitor is the granulocyte monocyte progenitor (GMP), the precursors to granulocytes and granulocytic myeloid-derived suppressor cells (gMDSCs). Immunologists have recently recognized that neutrophils play an important role in cancer immunity. Two subsets of tumor-associated neutrophils (TANs) can be recruited to the tumor microenvironment, as well as pre-metastatic niches [31]. Several subsets of TANs exist, some of which function in an antitumor capacity and others which promote tumor growth and development [25].

Hematopoiesis is canonically thought to occur within the bone marrow compartment, and its niche contains a variety of cells that support HSC retention, survival, and regulate function. Some of these supportive cells include mesenchymal stem cells, stromal cells, osteoblasts, adipocytes, CXCR12 abundant reticular cells (CAR cells), megakaryocytes, as well as sympathetic neuron Schwann cells. HSCs are retained within the bone marrow compartment via the CXCL12/CXCR4, CCL2/CCR2, and other chemokines axes. Myelopoiesis, specifically, is thought to occur within the central bone marrow niche, and a myeloid bias often occurs with increasing age [32,33].

Several transcription factors are known to play a role in promoting myeloid cell differentiation. These include CCAAT enhancer binding protein alpha (C/EBPα), PU.1, and GATA-1 [34]. Differentiation into myeloid cells occurs via a variety of cytokines and growth factors, such as stem cell factor (SCF) and thrombopoietin (TPO), as well as macrophage colony stimulating factor (M-CSF) and granulocyte macrophage (GM-CSF).

## 4. Impact of Solid Cancers on Dendritic Cell Differentiation

A series of studies from the late 1990s to the early 2000s demonstrated a significant interaction between tumors and dendritic cells (DCs) that progressed into an evaluation of myelopoiesis, as a whole. In two papers by Gabrilovich et al. in 1996, it was demonstrated that tumors can impair the differentiation of DCs from DC precursors [35,36]. Despite this, in an elegant series of experiments, they demonstrated that BM cells from tumor-bearing animals were able to be driven into differentiation into functional DCs when cultured in GM-CSF and IL-4 [36]. They were capable of presenting antigen to T cells in equal amounts to control samples. Therefore, they anticipated that tumors may be driving some of the defect in DC maturation and function. They then tested this by using tumor supernatants in a conditioned media experiment. Interestingly, mature DCs from the spleen did not have any defects in antigen presentation when in tumor supernatant-conditioned media. However, when they used BM cells, a heterogeneous population that includes myeloid progenitors, in a GM-CSF, IL-4, and 20% tumor supernatant culture, those progeny were functionally impaired in their ability to activate T cells. Therefore, tumor cell supernatants appeared to exert their influence on myeloid progenitors, but not so much on the mature myeloid effector cells.

In the next paper, they more specifically narrowed their analysis to specific cell subsets and specific soluble factors driving the connection between tumor and progenitors. Whereas before they used murine BM cells, here, they used CD34^+^ human cells. In repeating their supernatant transfer setup from before, they redemonstrated the phenomenon of tumor supernatant impact on functional DC maturation. They then determined that CD34^+^ cells cultured in tumor supernatants generated more progeny when compared to control cultures. When they analyzed this further, they determined there was a 2 to 3-fold decrease in mature DCs in the tumor culture groups, and that the progeny expressed less MHC II, had a reduced ability to take up soluble antigen, and were morphologically distinct from DCs. In a series of antibody blockade experiments, they identified VEGF as a key component within the broad milieu of tumor-released factors that impacted the differentiation of functional DCs [35]. While they did phenotype the cells generated by tumor supernatants with a number of markers, including CD33 and CD13, a broad phenotypic panel was somewhat lacking in these early studies. The implication of their findings, however, was that the tumor-released factors impacted an early stage hematopoietic or myeloid progenitor and did not impact more mature DCs.

In a paper a couple of years later, this mechanism was tied up in an elegantly performed in vivo study of Langerhans cells and their function in tumor-bearing hosts [37]. In total, the presence of tumors impaired the function of Langerhans cells, including their ability to home to lymph nodes and to activate T cells. In following up their studies on tumor-secreted factors, they hypothesized that in vivo tumors were secreting something that was impacting Langerhans cells. When they added anti-VEGF to the platform, Langerhans cells in tumor-bearing hosts were rescued in their functional ability, although limited in their combinatorial antitumor activity. The same group later studied 93 patients with breast, head and neck, and lung malignancies and recapitulated their preclinical findings [38]. Patients with solid tumors had fewer DCs in the periphery and were associated with an increase in immature myeloid cells. Interestingly, resection at least partially helped in reversing the hematopoietic derangements. In addition, they discovered that VEGF once again was a key driver of this phenomenon. These studies have multiple impacts. They insinuated that anti-VEGF therapy may enact its anticancer activity in more than just its impact on vasculature, while simultaneously demonstrating that tumors significantly regulate myelopoiesis for their benefit at an early developmental stage.

## 5. Impact of Solid Cancers on Myeloid Progenitors

In other papers from the same time period as those early DC differentiation studies [39,40,41], investigators were trying to determine what specific cells are promoted by solid tumors. Studies determined they were heterogeneous, and were globally termed immature cells (ImCs) from the myeloid lineage [39]. While specific progenitor populations were not yet understood at the level of detail they are now, there was an overall determination that one-third of ImCs were from the macrophage and DC lineage, and that two-thirds were earlier myeloid progenitors. Specific marker positivity of ImCs was <2% CD34^+^, 98% MHC-I, 30% CD115^+^CD11c^+^, 60% CD13^+^, and 20% expressed intracellular HLA-DR. While they phenotypically expressed those markers, functionally, ImCs were capable of impairing healthy DCs from activating T cells. In elegantly performed studies, sorting out the ImCs restored the function of DCs that were able to activate DCs adequately. Additionally, ImCs were capable of being restored themselves by simply driving their differentiation past the immature state by using all-trans retinoic acid and other molecules to polarize differentiation, including GM-CSF. In other studies at the time, GM-CSF from tumors was shown to promote a myeloid cell population that suppressed CD8^+^ T cells [41]. In those studies, this cell population was CD11b^+^Gr-1^+^, and did not express canonical DC markers but expressed some macrophage markers such as F4/80. Additionally, while GM-CSF drove the generation of these cells, the subsequent application of IL-4 with GM-CSF to these cells could also promote them to differentiate into mature antigen-presenting cells.

In concurrent studies at the time, STAT3 was implicated as a primary driver of tumor-mediated impairment of the immune system [42,43]. In those studies, tumor culture media was capable of driving STAT3 activation and downstream promotion of ImCs. When tumor factors were removed or STAT3 was inihibited, the ImC generation was reversed. With these studies, a potential mechanism of action of tumor-driven impaired DC generation and subsequent promotion of ImCs was demonstrated. Subsequent studies in more recent years have implicated the RORC1 pathway in tumor-mediated MDSC generation [44]. Specifically, when responding to GM-CSF, M-CSF, and G-CSF, the RORC1-expressing cells generate MDSCs and tumor-associated macrophages. When RORC1 is taken out of the system, tumor development is impaired and pro-inflammatory cells are restored. Multiple studies since then have redemonstrated the impact of tumor-mediated GM-CSF signaling on the generation of TME MDSCs [6,45].

After those initial studies showing the expansion of ImCs, many studies showed an expansion of MDSCs or other subtypes of early immunosuppressive myeloid cells in the peripheral blood of solid cancer-bearing hosts [46,47,48]. In 2005, it was demonstrated that MDSCs were expanded in the peripheral blood of renal cell carcinoma patients [47]. These studies showed elevated arginase activity in the peripheral blood, and subsequently discovered an expansion of gMDSCs that were CD11b^+^CD14^−^CD15^+^. When those cells were depleted, T cells were capable of expansion. Later studies in 2006 showed that, through T cell-activation cytokines, there was an expansion of CD11b^+^IL-4Rα^+^ inflammatory monocytes that were, in turn, capable of suppressing CD8^+^ T cells [48]. Additional studies even showed a potential link between therapy and expansion of MDSCs in the peripheral blood. In metastatic melanoma patients being treated with heat shock protein vaccines and GM-CSF, peripheral blood had expanded CD14^+^HLA-DR^−^cells in all patients [49]. Ultimately, MDSCs expansion in peripheral blood has been demonstrated in many tumor-bearing hosts.

In a seminal 2014 paper by Wu et al., many of these questions regarding which progenitors are affected by solid tumors were answered [50]. It was demonstrated that HSPCs in seven different types of cancer were increased in quantity and myeloid-biased [50]. The breadth of tumors included breast, esophageal, cervical, hepatocellular, gastrointestinal, lung, and ovarian tumors from 133 patients who were untreated and still in the throes of their native immune response to their cancer. They specifically identified an increase in HSCs, MPPs, and GMPs, while there was a decrease in CMPs. Interestingly, they also identified a positive correlation between the quantity of GMPs and increased stage of cancer. Mechanistically, they describe an expansion of GMPs in the blood of cancer patients that depended on IL-6, GM-CSF, and G-CSF to follow a granulocytic differentiation. Ultimately, these cells were driven into differentiation into MDSCs, the aforementioned cell type that often drives the TME into an immunoregulatory environment that turns off antitumor immune responses. By demonstrating this, Wu et al. added significant detail to an area of research on myeloid-bias in stem and progenitor cells in cancer patients.

The same group of researchers found that glutamine deprivation drives GM-CSF and G-CSF secretion by 4T1 breast tumor cells. They also found mice bearing 4T1 cells possess an expansion of GMPs within their splenic compartment relative to normal, nontumor-bearing mice. Additionally, they found CXCL12, the chemokine responsible for retaining HSPCs in the bone marrow compartment, to be significantly reduced in HSPCs from tumor-bearing mice relative to control mice. Using tumor supernatants, they found cKit+ precursors differentiate into a higher frequency of mMDSC and gMDSCs relative to control culture conditions. Finally, they stratified breast cancer patients based on expression of GM-CSF and found those with higher expression have worse overall survival relative to those with low GM-CSF expression. Overall, the authors suggest tumor-derived GM-CSF and G-CSF via glutamine deprivation drives myeloid cell expansion [51]. This followed on previous studies that already demonstrated a significant breast cancer-myelopoiesis axis of interaction [52,53].

## 6. Extramedullary Hematopoiesis

Extramedullary hematopoiesis (EMH), when hematopoiesis occurs outside of the bone marrow niche, is well characterized in embryonic development. However, EMH can also occur under inflammatory conditions such as atherosclerosis [54], sepsis [55], and infection [56]. Clinically, the most common sites of extramedullary hematopoiesis are the spleen and liver. In general, these studies have found an expansion of myeloid cells in the bone marrow and spleen relative to appropriate noninflammatory control groups. Wu et al. recently published an in depth review on the role of the spleen in cancer-induced myelopoiesis in mice [57], but we outline some seminal papers on EMH in solid tumors below.

Elegant studies by Cortez-Retamozo et al. have shown when spleens of tumor-bearing Kras^LSL−G12D/+^; p53^fl/fl^ (referred to as KP) mice are transplanted into nontumor-bearing mice, the donor-derived tumor-bearing splenocytes more effectively migrate to the tumor microenvironment relative to spleens from nontumor-bearing animals. In addition, they show tumor-bearing KP mice possess an expansion of lineage-CD11b^+^ cells in S/G2 phase, as well as an expansion of GMPs in tumor-bearing mice relative to nontumor-bearing controls. Finally, they show an expansion of splenic GMPs, monocytes, and neutrophils in mice with invasive cancer relative to control mice [58].

One group found an expansion of granulocyte colony-forming units in mice bearing hepatocellular carcinoma (HCC) relative to nontumor-bearing control mice. They also found higher expression of GM-CSF and CCR2 on lin^-^cKit^+^ Sca-1^hi^ (LSK). Furthermore, increased secretion of CCL2 in splenic stromal cells of HCC-bearing mice was also observed, suggesting the CCL2/CCR2 axis is important for splenic myelopoiesis. They determined that combining splenectomy with the immune checkpoint inhibitor (ICI), anti-PD-L1, provided a survival benefit relative to sham surgery control groups, and reduced LSK and GMP frequencies in the spleen. Finally, they found that, when HCC-bearing CCR2^−/−^ mice were treated with anti-PD-L1, a significant survival benefit was observed relative to control CCR2^+/+^ mice [59].

Levy et al. have specifically delineated the role of the spleen in mice bearing non-small cell lung cancer (NSCLC). They found tumor-bearing mice who had their spleens removed had reduced tumor growth relative to sham control mice when the removal was performed at more advanced disease stages. The authors also found a reduction in both CD11b^+^ Ly6C^+^ and CD11b^+^ Ly6C^+^ CCR2^+^ cells in splenectomized mice relative to sham control mice. Interestingly, they found mice that underwent a splenectomy had reduced CCL2 in the serum, relative to control mice. Finally, they found mice that received combinatorial splenectomy and a Gr-1 depleting antibody had reduced tumor growth relative to control mice, but that this effect could be lost when MDSCs were adoptively transferred, suggesting MDSCs were accumulating in the spleen [60].

Another group of researchers have characterized EMH in melanoma-bearing mice. They found moribund mice have larger spleens and increased overall splenic cell counts relative to non-tumor-bearing control mice. In addition, they determined melanoma-bearing mice possess an expansion of lin^-^, LSK, and myeloid precursor relative to nontumor control mice. Furthermore, they found an expansion of CMPs, GMPs, and MEPs in melanoma-bearing mice relative to control mice. Using melanoma conditioned media, the authors show IL-3 drives an expansion of lin^-^, LSKs, myeloid precursors, CMPs, and GMPs from whole BM cells. Finally, the authors administered anti-IL-3 to melanoma-bearing mice and analyzed the spleen immune cell populations, and found increased frequencies of lineage-, LSKs, and myeloid precursors relative to mice treated with isotype control. These results suggest melanoma induces splenic EMH and is driven by IL-3 signaling [61].

Allen et al. demonstrate the systemic immune profile, or immune macroenvironment, is altered throughout tumor progression [62]. Within the myeloid compartment, they describe an expansion of neutrophils and a reduction in eosinophils within the bone marrow, blood, and spleens of breast tumor-bearing mice. Importantly, they describe the alteration of peripheral immune profiles across a variety of murine solid tumor models and show animals that which undergo surgical resection can reset the immune composition of the spleen in a similar fashion to mice treated with G-CSF or IL-1b blockade.

Less is known about EMH in cancer in humans. EMH has been well studied in patients with hematological malignancies but has only recently been reviewed in those with solid tumors and no hematopathy [63]. A separate case report identified a patient with mismatch repair deficient (MMRd) colon cancer and EMH, with persistent rising levels of carcinoembryonic antigen (CEA) and a liver lesion while undergoing chemotherapy, both of which are usually consistent with tumor progression. Upon biopsy, it was revealed these lesions were sites of EMH and not sites of metastases [64]. Furthermore, Wu et al. determined patients with hepatocellular carcinoma, kidney, or pancreatic cancer had more CD133^+^ cells, a marker of human HSCs, as well as more CD11b^+^ cells relative to cirrhotic patients [59]. However, more studies evaluating the mechanism behind EMH in patients with solid tumors are needed.

## 7. Impact of Early Myeloid Cells on Cancer Progression and Metastasis

In a series of papers spanning from 2001 to the present, early hematopoietic cells have been described as promoting tumor growth and metastasis through specific VEGF-related mechanisms. In 2001, Lyden et al. described a role of BM-derived cells in promoting angiogenesis in tumors [65]. While the primary cell described was more of an early hematopoietic progenitor of endothelial cells, follow-up studies from the same group continued to demonstrate a specific interaction between the marrow and solid tumors. In follow-up work, Kaplan et al. described, in great detail, the role of BM-derived VEGFR-1^+^ cells in starting the set-up and promotion of pre-metastatic sites, or the sites at which future metastases will seed [66]. Through these studies, the role of BM-derived cells in native cancer immunity is thoroughly demonstrated. In subsequent studies from groups working on similar projects, the focus has shifted to more myeloid-biased cell populations. In the 2016 *Cancer Research* paper from Giles et al., it was demonstrated that solid tumors can significantly affect the bone marrow [67]. Specific findings included the tumor-mediated promotion of HSPC expansion in bone marrow through FLT3 ligand signaling. In addition, they found mobilization of HSPCs were mediated by the tumor. Perhaps most interestingly, they found that tumors mediated HSPC differentiation into myeloid cells with an immunosuppressive phenotype, including tumor-homing MDSCs, that ultimately were the drivers of pre-metastatic sites. In more clinical studies, they also demonstrated the same findings of Wu et al. in 2014. i.e., that cancer patients had an expansion of HSPCs in peripheral circulation and that that often correlated with advanced disease, including metastasis.

In a series of highly related studies, Qian et al. demonstrated, in 2011, that the CCL2-CCR2 axis is often at fault for the migration of early myeloid progenitors to tumor sites [68]. They found that CCL2-CCR2 mediated the migration of macrophage progenitors, which they call the Gr-1^+^ inflammatory monocyte, as well as driving extravasation, seeding, and growth of tumor cells. These effector myeloid cells described in the above studies are likely under the broad umbrella of MDSCs. While interesting and novel for the cancer environment, these findings have been reported elsewhere in other migration settings [69,70]. However, these studies demonstrate the key concept that tumors have communication with peripheral immunity and are drivers of myeloid progenitors that can, in turn, affect tumor seeding, growth, and, eventually, can impact their ability to be immunologically cleared. In more recent studies, MDSC expansion in the periphery has been noted to be driven not only by cancer presence, but also by lymphodepleting regimens that can impact ACT effectiveness [71,72]. These studies show the importance of not only tumor-mediated changes to myelopoiesis, but the impact of each individual treatment modality on myelopoiesis.

Solid cancer makes remarkable changes to the host myeloid compartments in the bone marrow, spleen, and peripheral blood. While many successful immunotherapies have focused on the adaptive arm of immunity, there is considerable potential in the myeloid compartment for reprograming tumor-induced dysmyelopoiesis (Figure 2).

## 8. Early Hematopoietic Cells as Immunotherapy

Given the significant immunomodulatory potential of early myeloid cells, the following discussion will cover some of the current strategies using HSPC transfers to overcome tumor-mediated derangements in myelopoiesis. However, there are a number of important targetable aspects of the mechanisms outlined above that deserve attention. Given the papers outlined above, the steps at which a potential immunotherapeutic intervention could be possible includes the following: 1. decreasing HSPC differentiation into MDSCs; 2. impairing the function of MDSCs in suppressing immunity; 3. increasing the differentiation of HSPCs and MDSCs into terminal effector DCs; 4. increasing the myeloid effector antigen-presenting function in tumors; 5. checkpoint inhibition of immunoregulatory molecule expression on myeloid effector cells; and 6. engineering chemokines/cytokines/other factors into myeloid effector cells (Figure 3). Specific targeted therapies that have been promising include the modulation of immunosuppressive myeloid cell generation in the TME [73]. Most of these mechanisms of action are focused on flipping the switch from an immunosuppressive to an immune-activating TME, or an M2 to an M1 phenotype [12,13,74]. One of the most promising is the development of CSF1R antagonists. These therapies are targeted against the primary mechanism of MDSC and TAM recruitment, activation, and immunosuppressive function that occurs through CSF1R signaling [74,75,76,77,78,79,80,81,82]. While considerable work has been performed, an optimal combinatorial platform for clinical success has not yet been developed [76]. The following discussion will provide an overview of strategies using HSPC transfers as immune-modulating therapies.

In a 2007 study from Wrezinski et al. [83], HSPCs were identified as key players in immunotherapy. In that study, mice bearing melanoma tumors were investigated during adoptive T cell therapy. Much of the relationship between solid cancer and HSCs was previously thought of as a simple rescue cell transfer. That is, HSCs were used for bone marrow rescue for myeloablated hosts that had their marrow depleted with chemotherapy or radiation. In addition, as we have identified in a number of studies in this review, many research findings identified HSPCs as a largely immunosuppressive cell type capable of driving further cancer progression and metastasis. In this 2007 study, however, they identified a role for HSCs in positively interacting with T cells from adoptive transfer and synergizing to generate antitumor responses. They found that lin^−^c-kit^+^ cells given with adoptive T cells actually enhanced the expansion and engraftment of T cells in the host. In addition, the effects of myeloablation were not the causative force for HSC-mediated T cell proliferation. In fact, the mechanism was instead tied to IL-7 and IL-15 in the myeloablated and treated melanoma-bearing hosts. In further follow-up experiments, they discovered that T cells just had to be present to provide this function, and their activation state was apparently not required.

While that study identified a role for HSCs in immunotherapy treatment platforms, it did not directly link the T cell activation to a terminal effector myeloid cell function. Instead, a series of papers published from 2013 to the present describe a more exact function of HSPCs and their progeny in enacting an antitumor function [84,85,86]. In Flores et al., from 2015 [84], HSCs were studied in the context of an immunotherapy platform that included polyclonal T cell therapy with dendritic cell vaccines to treat malignant gliomas. In these studies, HSPCs migrated to syngeneic brain tumors and chemotactically attracted adoptively transferred T cells to the TME. The mechanism responsible for this migration was the chemotactic gradient of HSPC-released MIP-1α in the TME that attracted adoptively-transferred T cells. This study identified HSPC-mediated T cell migration to the TME. However, the ultimate differentiation of that HSPC into terminal effector cells with distinct functions was not yet understood.

In a mechanistic study by Wildes et al. in 2018, the role of transferred HSPCs and their progeny during immunotherapy treatment was further elucidated [86]. The HSPC transfer was performed with adoptive cell therapy, but the HSPC-derived cells were tracked using DsRed. When the HSPCs migrated to the TME, they were still multipotent cells capable of rescuing other myeloablated hosts. Given the co-localization of HSPCs and T cells in the TME, the release of T cell cytokines was investigated as a potential source of influence on HSPC differentiation in the TME. Those experiments discovered that T cells in the TME release IFN-gamma, and that that IFN-gamma was the primary driver of HSPC differentiation into dendritic cells in the TME. This was shown through supernatant transfer experiments in which activated T cells in culture released soluble factors that drove HSPC differentiation. After knockout and antibody blockade experiments, this was determined to be largely due to IFN-gamma signaling. In vivo, this was redemonstrated with the use of IFN-gamma receptor knockout mouse-derived HSPCs that impaired the ability of HSPCs to differentiate into DCs in the TME. Functionally, the HSPC-derived DCs in the TME were capable of capturing and presenting tumor antigen to CD8^+^ T cells in the TME, thereby further activating them and continuing the feedback cycle. In that setting, the activated cells can have a strong role in influencing the differentiation patterns of the multipotent cells and can positively influence the often immunosuppressed environment of a cancer patient’s immune system. In that same study, MDSCs in the TME were actually supplanted by HSPC-derived DCs. In a further study on the mechanisms of HSPCs and their interaction with immunotherapy, Flores et al. investigated HSPCs with PD-1 checkpoint blockade [85]. These studies determined that there was also antitumor efficacy with HSPCs and PD-1 checkpoint blockade. Interestingly, the CCR2^+^ fraction of HSPCs was specifically responsible for the majority of antitumor activity. In addition, these CCR2^+^ cells were the population that differentiated into DCs in the TME and presented antigen to CD8^+^ T cells in the TME.

The major takeaway of these studies was that adoptive T cell therapy with HSPC transplant or PD-1 blockade with HSPC transplant could cure 30% of animals with recalcitrant brain tumors. However, the mechanistic understandings derived from them show a distinct role for the combinatorially and temporally synchronized infusion of activated immune cells with multipotent progenitor cells. This allows for the potential for modulation of the dysregulated myelopoiesis that often occurs natively in tumor-bearing hosts. Future studies in this experimental system will investigate the ability of immunotherapy to modulate the native dysregulated myelopoiesis that occurs in tumor-bearing hosts.

In a very recent study from Kaczanowska et al., they conceptually advance the idea of an early hematopoietic cell transfer through the use of genetically engineered myeloid cells (GEMys) [87]. In this paper, the therapeutic intervention was altering myeloid cells themselves to modulate the tumor microenvironment, instead of relying on other cells to drive their differentiation and functional program [87]. This paper from the Kaplan group follows on their previous work in pre-metastatic niches that was published within the last 15 years. Their prior work showed that early myeloid cells migrate to distant sites to set up an immunosuppressive microenvironment that turns off immunity and facilitates metastases. Here, they are reversing it with IL-12 derived from GEMys. For this study, they use HSPCs derived from the bone marrow to genetically engineer them to produce IL-12. Then, when these HSPCs with IL-12 constructs migrate to pre-metastatic niches, they are able to turn on immunity, prevent immunosuppression, activate T cells, induce IFN-gamma release, and limit metastases and improve survival. This study is a major advance in the ability to use HSPCs as a modifiable immunotherapy for solid cancers, which has been tried with varying success in other settings [23].

## 9. Conclusions

HSPCs and, specifically, myeloid cells, are known to modulate immunity in infections and hematologic malignancies. Less, however, is known about HSPCs in the context of solid tumors, especially outside the TME. This review provides an extensive overview of how myeloid cells in the periphery are affected by solid tumors. We discuss how solid tumors are capable of impairing DC generation and function, both of which are crucial for generating a long-lasting antitumor immune response. Furthermore, we review how myeloid progenitors are altered in cancer and are expanded via EMH in patients and murine models of solid tumors. We also examine the role by which myeloid progenitors promote metastases and tumor progression.

Although the exact mechanism by which tumors are able to hijack the hematopoietic compartment to promote immune suppression and metastases is likely multifactorial, several cytokines, including GM-CSF, M-CSF, and G-CSF, have been shown to promote the development of these myeloid progenitors. Future studies are needed to identify the ways by which cancer alters myelopoiesis at the genetic level, and whether these mechanisms are cell intrinsic.

Interestingly, our group and others have shown HSPCs are able of enhancing the antitumor effect that is seen in several forms of immunotherapy. Consistent with the above, the precise mechanism is yet to be elucidated; however, immunotherapy seems to be capable of redirecting differentiation of these HSPCs to immune cells that promote a survival benefit in tumor-bearing mice. We are hopeful that, within the coming years, combinatorial approaches using HSPCs and immunotherapy will provide promising antitumor results across a variety of cancer types.

## Figures and Tables

**Figure 1 cells-10-00968-f001:**
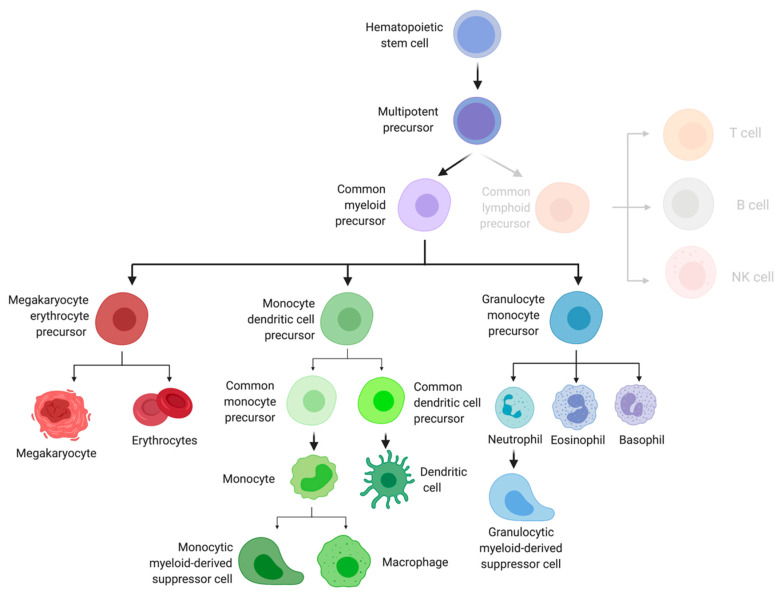
Myeloid differentiation flowchart. Image created with biorender.com (accessed on 25 March 2021).

**Figure 2 cells-10-00968-f002:**
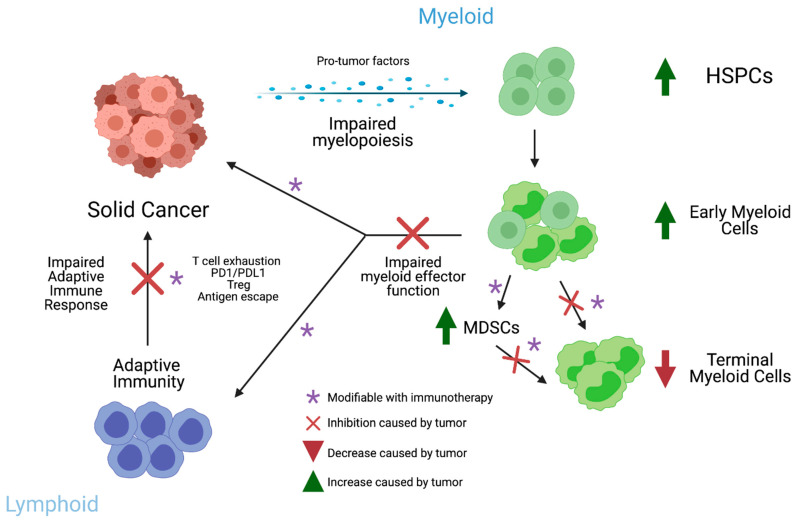
Schematic of interaction between myelopoiesis and cancer immunity. Image created using biorender.com (accessed on 25 March 2021).

**Figure 3 cells-10-00968-f003:**
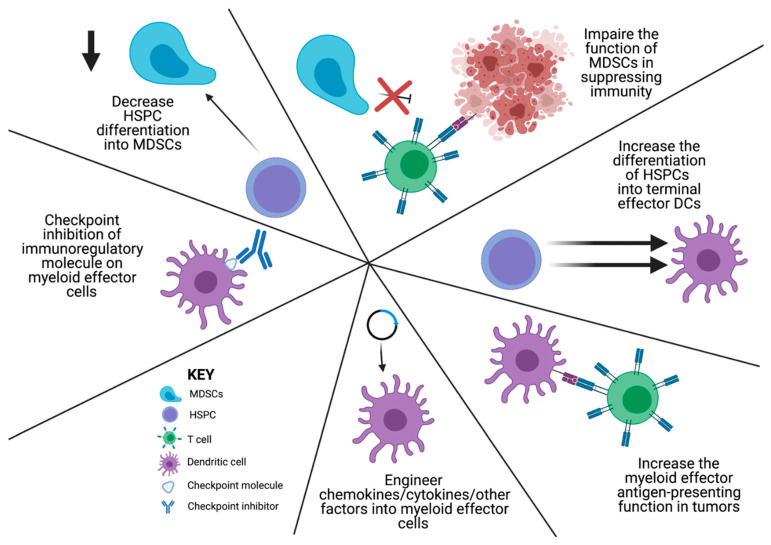
Schematic of the strategies to overcome cancer-mediated dysregulated myelopoiesis. Image created using biorender.com (accessed on 25 March 2021).

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
