# Peer review of "Myelopoiesis during Solid Cancers and Strategies for Immunotherapy"

_cells, 2021, doi:10.3390/cells10050968_

Round 1

Reviewer 1 Report

The manuscript nicely discusses the recent findings of interaction between myeloid calls and cancer immunity, especially myelopoiesis modulated by tumor cells in solid cancers. It also summarizes the mechanisms and potential of myeloid progenitor cells as immunotherapy.   

Several comments are made to help improve the manuscript.

  1. The title might be better to change to “Myelopoiesis in solid cancers and its modulation for therapeutic application” to reflect the full picture.
  2. Figure 1 is not specific and not necessary, since it can be found in the related textbook. The figure could be replace with a schema showing the impact of tumor derived factors on myelopoiesis.
  3. The potential intervention strategies of modulating myeloid cells for immunotherapy are better presented in a schema figure or to be integrated with Figure 2.

Minor ones:

  1. “control patients” needs to be changed to “control mice” on the line 235;
  2. “similar groups” on the line 289 should be corrected to “ the groups working on similar projects”;
  3. “furthering” should be changed to “further” on the line 368;
  4. “combinatorial” should be corrected to “combinatorially” on the line 382.

Author Response

Reviewer 1

The manuscript nicely discusses the recent findings of interaction between myeloid calls and cancer immunity, especially myelopoiesis modulated by tumor cells in solid cancers. It also summarizes the mechanisms and potential of myeloid progenitor cells as immunotherapy. Several comments are made to help improve the manuscript.

  1. The title might be better to change to “Myelopoiesis in solid cancers and its modulation for therapeutic application” to reflect the full picture.

Thank you for the suggestion. We agree that something similar to your suggested title would be more suitable and representative of the manuscript. We have changed it accordingly on lines 2-3.

  1. Figure 1 is not specific and not necessary, since it can be found in the related textbook. The figure could be replace with a schema showing the impact of tumor derived factors on myelopoiesis.

While figure 1 has components that are found elsewhere, we think a fundamental grasp of the myelopoiesis flow is required for understanding many of the topics in the manuscript. For the uninitiated reader, we think that even a cursory glance at figure 1 would greatly enhance their understanding of the rest of the concepts. Whereas is this was outsourced to another manuscript, the reader may not dig far enough to find that information. What we have done is added a new figure 3 that details the specific strategies that could be used to overcome dysregulated myelopoiesis. Figure 3 is on lines 481-482.

  1. The potential intervention strategies of modulating myeloid cells for immunotherapy are better presented in a schema figure or to be integrated with Figure 2.

Thank you for this suggestion. Based on your suggestion, we have provided an additional figure, figure 3, that covers some of the major therapeutic interventions that are discussed in the manuscript. Figure 3 is on lines 481-482.

Minor ones:

  1. “control patients” needs to be changed to “control mice” on the line 235;

This has been changed on line 342.

  1. “similar groups” on the line 289 should be corrected to “ the groups working on similar projects”;

This has been changed on line 408.

  1. “furthering” should be changed to “further” on the line 368;

This has been changed on line 531.

  1. “combinatorial” should be corrected to “combinatorially” on the line 382.

This has been changed on line 545.

Reviewer 2 Report

The authors summarize normal myeloid development and then review the effects of the presence of solid tumors on dendritic cell and myeloid cell development, focusing on blood, marrow, and extra-medullary hematopoiesis.  They then discuss impact of immature myeloid-derived suppressor cells on cancer, followed by discussion of use of early hematopoietic cells as immunotherapy (the latter an area of particular expertise of the authors).  Overall, this review adds a novel perspective focused on the effects of cancer on myeloid development outside of the tumor and therapeutic interventions using hematopoietic progenitors.

Dendritic cells are of several varieties, including some that are tumor-associated, e.g. CD103+ (papers by M. Merad).  The authors might describe this diversity and in reviewing the literature make reference to the subsets affected.

Much effort has been made to modulate the phenotypes of MDSCs, e.g. to shift their phenotypes from M2 to pro-inflammatory M1. For example, targeting CSF1R.  Perhaps the authors can acknowledge these efforts and cite recent reviews on this area at the beginning of the last section focused on early hematopoietic cells as therapy. 

Additional minor comments:

line 113:  change C/EBPepsilon to C/EBPalpha (which plays a more key role in early myeloid lineage specification)

line 115:  change thyroid peroxidase to thrombopoietin

lines 222/223: change "lymph node, liver, and kidney" to "spleen and liver"

line 310: change "5" to "6" 

Author Response

Reviewer 2

The authors summarize normal myeloid development and then review the effects of the presence of solid tumors on dendritic cell and myeloid cell development, focusing on blood, marrow, and extra-medullary hematopoiesis.  They then discuss impact of immature myeloid-derived suppressor cells on cancer, followed by discussion of use of early hematopoietic cells as immunotherapy (the latter an area of particular expertise of the authors).  Overall, this review adds a novel perspective focused on the effects of cancer on myeloid development outside of the tumor and therapeutic interventions using hematopoietic progenitors.

We are glad to hear you found that this review added a novel perspective.

Dendritic cells are of several varieties, including some that are tumor-associated, e.g. CD103+ (papers by M. Merad).  The authors might describe this diversity and in reviewing the literature make reference to the subsets affected.

We have added some details on CD103+ DCs from lines 55-57 and 78-89.

Much effort has been made to modulate the phenotypes of MDSCs, e.g. to shift their phenotypes from M2 to pro-inflammatory M1. For example, targeting CSF1R.  Perhaps the authors can acknowledge these efforts and cite recent reviews on this area at the beginning of the last section focused on early hematopoietic cells as therapy. 

Thank you for this suggestion. We have added in some text and citations acknowledging this area of investigation. These changes are on lines 454-464.

Additional minor comments:

line 113:  change C/EBPepsilon to C/EBPalpha (which plays a more key role in early myeloid lineage specification)

This has been changed on line 160.

line 115:  change thyroid peroxidase to thrombopoietin

This has been changed on line 162.

lines 222/223: change "lymph node, liver, and kidney" to "spleen and liver"

This has been changed on line 330.

line 310: change "5" to "6" 

There were a few numbering issues for the headings after edits that were corrected on lines 72, 108, 164, 219, 298, 397, and 443.

Reviewer 3 Report

Myelopoiesis during solid cancers, I don’t consider it is a good review’s title.

Due to as the author mentioned the purpose of this review is to delineate exactly what is known about how solid cancers impact native myelopoiesis. To achieve this, we will sometimes describe findings belonging to the broad category of hematopoietic stem and progenitor cells (HSPCs), a heterogeneous population of early myeloid progenitors and stem cells.

Speaking about title on any review it must be the same which gives clear direction to the review of any products which you are giving, However on making titles please be sure that the information provided by you should be correct.

Incorporate important keywords Consider what about your article will be most interesting to your audience: Most readers come to an article from a search engine, so take some time and include the important ones in your title!

Keywords: cancer; solid cancer; I suggest the author just list solid cancer in the Keywords.

Figure 1. Myeloid differentiation flowchart. Need revise, let it more clearly.

Line 79, the short-term HSC (stHSC) and multipotent progenitor (MPP)

Here, I suggest the author talk more about stHSC and MPP.

Line 121, tumors can impair the differentiation of DCs from DC precursors

I suggest the author cite more references as below.

Dendritic cells in cancer immunology and immunotherapy Nature Reviews Immunology volume 20, pages7–24(2020)

Dendritic cells in cancer: the role revisited. Curr Opin Immunol. 2017 Apr; 45: 43–51.PMID: 28192720

For the section 4. Impact of solid cancers on myeloid progenitors

This part is the key core of the review, the author should discuss more about it, many publications not cited, such as below.

Myelopoiesis, metabolism and therapy: a crucial crossroads in cancer progression

Cell Stress. 2019 Sep; 3(9): 284–294. PMID: 31535085

Myelopoiesis in the Context of Innate Immunity.J Innate Immun 2018;10:365–372

Reactive myelopoiesis and the onset of myeloid-mediated immune suppression: Implications for adoptive cell therapy Cellular Immunology Volume 361, March 2021, 104277

For a review on this topic, the totally references must more than 100, I suggest the author cite more papers to support his ideas and also could show a more convincing prospective for future research.

“several chemokines and growth factors have shown to promote the migration and development of these myeloid progenitors. Future studies are needed to identify the way by which cancer alters myelopoiesis at the genetic level and if these mechanisms are cell intrinsic.”  Several chemokines and growth factors , please list more details in here about them.

Author Response

Reviewer 3

Myelopoiesis during solid cancers, I don’t consider it is a good review’s title. Due to as the author mentioned the purpose of this review is to delineate exactly what is known about how solid cancers impact native myelopoiesis. To achieve this, we will sometimes describe findings belonging to the broad category of hematopoietic stem and progenitor cells (HSPCs), a heterogeneous population of early myeloid progenitors and stem cells. Speaking about title on any review it must be the same which gives clear direction to the review of any products which you are giving, However on making titles please be sure that the information provided by you should be correct.

We have changed the title to accommodate your comment and reviewer 1’s comment on lines 2-3.

Incorporate important keywords Consider what about your article will be most interesting to your audience: Most readers come to an article from a search engine, so take some time and include the important ones in your title! Keywords: cancer; solid cancer; I suggest the author just list solid cancer in the Keywords.

We have re-considered our keywords and still find them comprehensive and noteworthy. Per your recommendation we have removed cancer as a keyword (line 27), and we have updated the title of the manuscript to highlight keywords and better describe the review (lines 2-3).

Figure 1. Myeloid differentiation flowchart. Need revise, let it more clearly.

We had difficulty understanding what specifically needed to be changed on the differentiation flowchart based on your comment. We reviewed the differentiation figure and it appears clear to us. However, we have provided an additional figure, figure 3, to cover the strategies to overcome dysregulated myelopoiesis on lines 481-482.

Line 79, the short-term HSC (stHSC) and multipotent progenitor (MPP)

Here, I suggest the author talk more about stHSC and MPP.

Thank you for the suggestion. We have added in some detail on this topic in lines 123-125.

Line 121, tumors can impair the differentiation of DCs from DC precursors

I suggest the author cite more references as below.

Dendritic cells in cancer immunology and immunotherapy Nature Reviews Immunology volume 20, pages7–24(2020)

Dendritic cells in cancer: the role revisited. Curr Opin Immunol. 2017 Apr; 45: 43–51.PMID: 28192720

For the section 4. Impact of solid cancers on myeloid progenitors This part is the key core of the review, the author should discuss more about it, many publications not cited, such as below. Myelopoiesis, metabolism and therapy: a crucial crossroads in cancer progression

Cell Stress. 2019 Sep; 3(9): 284–294. PMID: 31535085. Myelopoiesis in the Context of Innate Immunity.J Innate Immun 2018;10:365–372. Reactive myelopoiesis and the onset of myeloid-mediated immune suppression: Implications for adoptive cell therapy Cellular Immunology Volume 361, March 2021, 104277

Thank you for the suggestion. While we find reviews helpful for finding primary articles, we have tried to not cite other reviews but instead focus on primary manuscripts for detailed discussions. We did add in text covering a couple primary manuscripts covered in the reviews you recommended and recommended by reviewer 2. We have additionally cited reviews in areas where we did not go into detail. In total, we have cited 37 more papers per your recommendation. These changes are throughout the manuscript. Specifically relating to your topics or interest these are located in lines 72-107, 233-269, and 428-433.

For a review on this topic, the totally references must more than 100, I suggest the author cite more papers to support his ideas and also could show a more convincing prospective for future research.

Several chemokines and growth factors , please list more details in here about them.

We have cited 37 new citations including papers you recommended. We have also updated the manuscript to have an additional figure highlighting the strategies that can be employed for future research (figure 3). We have also updated the conclusions with brief detail (577-578). We have limited the amount of details included in the conclusion and instead kept those details in the main body of the manuscript where specific papers are discussed.

Round 2

Reviewer 3 Report

Most concerns were answered well now.